# A One Health Perspective on the Resurgence of Flea-Borne Typhus in Texas in the 21st Century: Part 1: The Bacteria, the Cat Flea, Urbanization, and Climate Change

**DOI:** 10.3390/pathogens14020154

**Published:** 2025-02-05

**Authors:** Gregory M. Anstead

**Affiliations:** 1Division of Infectious Diseases, Medical Service, South Texas Veterans Health Care System, 7400 Merton Minter Blvd, San Antonio, TX 78229, USA; anstead@uthscsa.edu; 2Division of Infectious Diseases, Depatment of Medicine, University of Texas Health San Antonio, 7703 Floyd Curl Drive, San Antonio, TX 78229, USA

**Keywords:** flea-borne typhus, rickettsiae, opossum, cat, dog, *Ctenocephalides felis*, climate change, urbanization

## Abstract

Flea-borne typhus (FBT), due to *Rickettsia typhi* and *R. felis*, is an infection typically causing fever, headache, rash, hepatitis, and thrombocytopenia. About one quarter of patients suffer pulmonary, neurologic, hematologic, renal, hepatic, cardiac, ocular or other complications. In the 21st century, the incidence of FBT has increased in both Texas and California compared to the 1990s. In this paper, county-level epidemiological data for the number of cases of FBT occurring in Texas for two decades, 1990–1999 and 2010–2019, were compared with respect to county of residence, urbanization, and climatic region. Human population growth in Texas has promoted FBT by increased urbanization and the abundance of pet dogs and cats, stray/feral dogs and cats, and opossums. Increasing temperatures in Texas in the new millennium have increased the flea-borne transmission of FBT by promoting host infestation and flea feeding and defecation, accelerating the flea life cycle, and increasing rickettsial replication within the flea. Increased numbers of opossums and stray cats and dogs in the urban/suburban landscape have increased the risk of flea transfer to humans and their pets.

## 1. Introduction

Flea-borne typhus (FBT) is an infection caused by the bacteria *Rickettsia typhi* and *R. felis*. It is typically an acute undifferentiated febrile illness, but about one quarter of patients suffer respiratory, neurologic, renal, hepatic, cardiac, ocular, or other complications [1,2]. The infection is transmitted to humans by a flea bite; by the inoculation of a bite site, a skin abrasion, or mucous membranes with feces from fleas infected with these rickettsiae; or by the inhalation of infected flea feces [3,4,5]. In the new millennium, the incidence of FBT has increased in both Texas (TX) and California (CA) [6,7,8]. During 1990–1999, 307 cases of FBT were reported in TX [9]; in the decade 2010 to 2019, there were 3750 cases [6], which was a 12-fold increase.

The One Health concept focuses on issues at the intersection of human and animal health and the environment [10]. This manuscript is the first part of a two-part series that integrates the current knowledge of the characteristics of the pathogens and vectors of FBT, the flea host mammals, the environment, and human population trends to derive a One Health model to explain the increase in the number of FBT cases in TX that has occurred in the new millennium. In this schema, the principal driver of the increased number of FBT cases in TX is ecosystem disruption due to human population growth/urbanization and climate change. The current epidemiologic trends of FBT in TX are dependent on two versatile opportunists, the cat flea *Ctenocephalides felis* and the Virginia opossum *Didelphis virginiana*. Furthermore, increasing populations of cats (*Felis catus*) and dogs (*Canis familiaris*), including stray and feral animals, may also be affecting the epidemiologic trends [11]. The increasing populations of these mammals are being driven by human population growth. Greater mammalian host abundance may increase cat flea populations, which may also be increased by climate change. In the second part of the series, the natural histories of the host mammals and their relationship to the epidemiology of FBT will be discussed.

## 2. Methods

Since 1946, the Annual Summary of Notifiable Diseases published by the Texas Department of Health (earlier agency), or the Texas Department of State Health Services (TDSHS, the current agency), has included data on FBT [12]. In this paper, county-level epidemiological data for the number of cases of FBT occurring in TX for two decades, 1990–1999 [9] and 2010–2019 [6], were compared with respect to county of residence, urbanization, and climatic region. County-level data were obtained from the TDSHS. Temperature data were acquired using the Prism Climate Group database [13]. The choropleth map was prepared using Datawrappper [14]. Populations of TX counties for the indicated years were obtained from US Census data. Historically, most diagnoses of FBT were based on the serologic testing of serum of symptomatic patients [15]. More recently, the next-generation sequencing technique of whole blood for the detection of the presence of rickettsiae has become commercially available [16,17]. Texas law requires that healthcare providers, hospitals, and laboratories inform the TDSHS if they suspect that someone has a notifiable condition, which includes FBT [18].

## 3. Historical Epidemiology of Flea-Borne Typhus in Texas, 1923–1999: The Effects of Changing Agricultural Practices and Rodent/Flea Control Measures

An episystem is defined as “the set of biological and environmental elements, as well as the epidemiological aspects of these diseases in defined geographic and temporal scales” [19]. In this paper, the geographic space is the state of TX, and the time scale that will be discussed is 1930 to 2019. The abiotic, biotic, and anthropogenic components of this episystem are listed in Table 1.

In TX in 1923, only five cases of FBT were reported. By 1933, the number of recorded cases had increased to 399 [20], which was likely due to better case ascertainment [21]. In the 1930s, many TX farmers made a transition from cotton cultivation to that of peanuts [22]. The advent of World War II and the Japanese capture of coconut-producing islands of the Pacific necessitated a greater US production of peanut oil because it replaced coconut oil for the production of glycerin, which was required in the manufacture of munitions. Spurred by the US Department of Agriculture, TX farmers tripled their peanut production during the war [23]. Texas ranked second among the states in acres planted in peanuts for the decade 1938–1948 [22].

During the 1930s and 1940s, the dominant reservoir animals for FBT were the brown and black rats, *Rattus norvegicus* and *R. rattus*, respectively, and the main vector was the Oriental rat flea, *Xenopsylla cheopis* [24,25]. Peanut cultivation in the American South in the 1930s and 1940s promoted rat proliferation, which likely contributed to an increase the number of cases of FBT [21]. Rat proliferation was not restricted only to areas of peanut cultivation because there were peanut processing facilities located throughout TX in the 1940s [22]. The annual number of cases of FBT peaked in TX in 1945 at 1844 cases [26]. By 1946, 180 of the 254 TX counties had reported cases [27] (Figure 1 and Figure 2). Other factors in the rise of FBT in Texas and the American South in the late 1930s and early 1940s have been described elsewhere [21].

Starting in 1945, there was a massive public health campaign to eradicate rats and their fleas in the American South. The incidence of FBT in TX in a single decade reached its nadir in the 1960s with only 246 cases reported from 1960 through 1969. However, by the late 1960s in Texas, FBT was undergoing a transition from the Oriental rat flea as the major vector to the cat flea [7,28].

From 1980 to 1984, there were 200 cases of FBT reported in TX with four south Texas counties dominating: two adjacent counties along the Gulf Coast (Nueces Co. with 57 cases and Kleberg Co. with 9 cases) and two counties in the Rio Grande Valley (Hidaldo Co., 39 cases, and Cameron Co., to the east, 22 cases) (see Figure 3) [29].

From 1990 to 1999, there were 307 cases of FBT in Texas [9] (Figure 4), occurring in 28 counties. Officially recorded cases of FBT in TX reached a nadir of nine cases in 1994 [9]. The majority of FBT cases in TX in the 1990s occurred south of the 66 °F isotherm (the isotherm is based on weather from 1981 to 2010) [30], which is the area climatically most favorable for FBT transmission. I refer to this area of higher FBT endemicity as the Lower Rio Grande–Texas Gulf Coast FBT Refugium; of the 307 total cases in this decade, 88.9% of cases occurred in the Refugium’s 13 counties and only 34 cases were seen outside the Refugium (11.1%). Another seven cases occurred in San Patricio County, adjacent to the 66 °F isotherm. If data from 1980–1984 are re-examined, 145 of the total 200 cases (72.5%) were reported from the Refugium counties (Figure 3) [29].

## 4. Typhus in Texas in the New Millenium: The 2003–2013 Texas DSHS Study

The first published observation of a dramatic increase in FBT cases in TX in the new millennium was the 2003–2013 Texas DSHS Study. During that period, a total of 1762 Typhus Group Rickettsioses (TGR) cases were reported to the TDSHS. The number of cases per year varied from 27 in 2003 to 222 in 2013. An average of 102 cases were reported annually during 2003–2007, which was less than half of the average number of annual cases (209) reported during the latter half of the period (2008–2013). Overall, TGR cases peaked in June and July; however, in south TX (≤28° N latitude), peaks occurred both in June/July and the other in December/January. The reason for this bimodal distribution of cases in south Texas was uncertain. Geographic expansion of FBT within TX also occurred during the 2003–2013 period. In 2003, cases were reported from nine counties in south Texas, but by 2013, cases had been reported from 41 counties, primarily in south and central TX [31]. The researchers were unable to discern the reasons for the increasing number of cases of TGR in TX from 2003 to 2013; they doubted that greater case ascertainment and reporting was the cause. The rising number of cases of TGR in TX is even more apparent if we look back to the 1990s and compare to more recent data. There were 307 cases in TX in the decade 1990 to 1999. This was an average of 30.7 per year, with 26 counties reporting cases; 82% of the cases occurred in two Lower Rio Grande Valley counties and one Gulf Coast county (Hidalgo, Cameron, and Nueces counties; 113, 21, and 117 cases, respectively) [9]. The TDSHS has released county-level FBT data from 2010 to 2019 [6]; in this manuscript, these data were analyzed for geographic patterns. In this ten-year period, there were 3750 cases (an average of 375 cases per year) reported from 91 TX counties. The purpose of the current manuscript is to derive a One Health model to account for this increase in FBT cases in TX in the new millennium by directly comparing 1990–1999 to 2010–2019.

## 5. Building a One Health Model: The Epidemiology of Typhus in Texas, 2010–2019: The Effect of the Population Growth of Humans and Their Pets

The human population of TX is increasing at a rapid rate. In 1990, the population of the Lone Star State was 17 million; in 2019, the population stood at 29 million (US Census), which is a 70.7% increase. The urban population of TX grew even more in this period, constituting 83%; thus, the proportion of Texans living in a metropolitan area increased from 83.4 to 89.4% (Table 2).

The populations of the six counties with the highest number of FBT cases (Bexar, Cameron, Harris, Hidalgo, Nueces, and Travis) in the decade 2010–2019 increased 67.1%, 62.7%, 66.3%, 126%, 24.2,% and 121%, from 1990 to 2019, respectively (Table 2). The rate of population growth of Hidalgo and Travis counties exceeded that of TX as a whole. Human population growth results in further urbanization/suburbanization with effects on FBT reservoir/flea host populations and local climate (which will be described below).

A TX county is defined as urban if the population is greater than 50,000 [33]. From 2010 to 2019, 3394 out of 3750 total cases of FBT (90.5%) occurred in counties classified as urban (Figure 5). Compared to 1990–1999, cases of FBT have increased markedly in Bexar, Cameron, Harris, Hidalgo, Nueces, and Travis counties in the period 2010–2019 (Table 3). These six urban counties had 2824 cases during 2010–2019, which comprised 75% of all cases in TX for that period. The increase in FBT cases in these growing urban counties far exceeded the number expected from human population growth alone. Likewise, in Orange Co., CA, cases of FBT were found in highly populated urban areas in the northern half of the county with no cases recorded in the less urbanized south [34].

A choropleth map (Figure 6) illustrates the relative number of cases for each county in Texas during 2010–2019. Clusters of high-case load counties are apparent in North Texas (Tarrant (93 cases) > Dallas (50)); central TX (Travis (239) > Williamson (16) > Hays (12)); upper coast Texas (Harris (252) > Galveston (99) > Brazoria (26) > Montgomery (20)); the Coastal Bend (Nueces (580) > San Patricio (58) > Kleberg (38) > Jim Wells (31) > Bee (19) > Victoria (12) > Refugio (11)) and the Rio Grande Valley (Hidalgo (963) > Cameron (368) > Webb (76) > Starr (47) > Willacy (33) > Zapata (12)).

Humans are accompanied by their companion animals, which may act as reservoirs for FBT and blood meal and transport hosts for fleas. In 2014, it was estimated that there were 7,163,000 pet dogs and 5,565,000 pet cats in TX [35]. At that time, the human population of TX was 27 million. Thus, there were 0.27 dogs and 0.21 cats per Texan. Assuming the popularity of dog and cat ownership has not significantly changed from 1990 to 2019, the estimated number of pet cats and dogs can be compared for the two time periods 1990–1999 and 2010–2019 (Table 4 and Table 5). There are no reliable estimates of the number of feral dogs in the U.S., but they are considered to be ubiquitous [36]. In 2009, Bexar County (TX) was estimated to have 100,000 stray dogs [37]; at that time, the human population of the county was 1.65 million (US census), giving a 17:1 human to stray dog ratio. If other Texas urban counties are similar, the stray and total dog populations can be estimated for the six Texas counties with the highest number of FBT cases (Table 4).

Likewise, in 2014, it was estimated that there were 5,565,000 pet cats in TX [35], averaging 0.21 cats per Texan. The number of stray and feral cats in the USA is also not known. Estimates are one-third to about equal of the owned cat population [38,39]. In warmer climates, there may be higher numbers of free-roaming cats, because in areas with mild winters, females may produce up to three litters per year [38]. If a conservative estimate of 40% of the pet cat population is used, this would be approximately 2.4 million free-roaming cats in TX in 2019. Table 5 provides an estimated number of stray and feral cats in TX and the six counties with highest number of FBT cases.

Thus, the total dog and cat populations of the top six Texas counties for FBT cases are estimated to be 3.06 million and 2.74 million, respectively for the years 2014/2015 (midpoint of the decade) compared to 1.99 million and 1.78 million, respectively, for the years 1994/1995 (midpoint of that decade). Thus, there were about 1 million additional dogs and 1 million additional cats in the six high incidence counties in the 2010s compared to the 1990s. This would have increased the number of potential reservoirs for FBT and sources of blood meals for adult and larval flea survival and provided an additional armada of transport hosts for fleas to infiltrate human yards and households.

In addition to its urban predominance, FBT also has a distinct climatic distribution in the state of TX. Climatic conditions may affect the rapidity of the flea’s life cycle; the survival of each of the stages of the life cycle; flea mating, feeding, and defecation rates; the extent of flea colonization with endosymbiotic organisms; the percent of reservoir animals infested; and the flea burden (these factors will be discussed in detail in this paper and the subsequent paper in the series). In the 1940s in the USA, FBT was most prevalent in areas where the average January temperature was above 40 °F and the average relative humidity in July at noon is above 37% [40]; these were the conditions most favorable for the survival of the Oriental rat flea (*Xenopsylla cheopis*), which was the most significant vector at that time [41]. Seasonally, the lowest numbers of FBT cases were reported in the fall and winter and the greatest in July through September. Except for 1940, the number of cases of FBT reported in the USA increased each year since recognition of the disease in the early 1920s until 1945 [21]. The decrease in cases in 1940 may have been due to an unusually cold January in that year. There was no association between precipitation and the incidence of FBT [42]. The main vector in TX is now the cat flea, but climatic factors will also likely affect the current distribution of FBT in the state.

Texas has a south-to-north gradient of decreasing mean annual temperature and increasing seasonal temperature variation and an east-to-west gradient of decreasing precipitation [30,43]. Based on the temperature, precipitation, humidity, vegetational characteristics, and seasonal weather changes, the National Climatic Data Center divides TX into 10 climate zones (Figure 7; Table 6) [30]. Flea-borne typhus cases in TX in the period 2010 to 2019 show a very distinct distribution in these climate zones. In the northern “cooler” half of TX, proceeding west to east from areas of lower precipitation to higher are the following divisions: (1) High Plains; (2) Low Rolling Plains; (3) Cross Timbers; and (4) Piney Woods. This northern set of climate divisions had only 229 reported cases of FBT during 2010–2019, constituting only 6.1% of the total TX cases (Table 6 and Figure 7). The divisions comprising the central belt across TX are, from west to east, (5) Trans Pecos; (6) the Edwards Plateau; (7) Post-Oak Savannah; and (8) Gulf Coastal Plains. These four divisions had 1, 56, 1465, and 408 cases of FBT in the period 2010 to 2019, respectively, amounting to 51.4% of the total. The two most southerly climatic divisions, from west to east, are the South Texas Plains and Lower Rio Grande Valley with 227 and 1364 cases, respectively (42.4% of the total). Thus, TX FBT cases are focused in the Post-Oak Savannah and Lower Rio Grande Valley climate divisions (1465 and 1364 cases, respectively), which account for 75.5% of the total cases (Figure 6). The Gulf Coastal Plains and South Texas Plains account for another 16.9% of the total cases (for average temperature and rainfall data for each climate region, see [30]). Compared to 1990–1999, in 2010–2019, FBT increased in eight of the ten climate regions, especially in the Gulf Coastal Plains and Cross-Timbers (136- and 49.5-fold, respectively).

From the 1960s through the 1990s, FBT mostly retreated to its subtropical refugium deep in south TX (see Figure 3 for 1980–1984 and Figure 4 for 1990–1999), with the majority of cases occurring south of the 28th parallel, approximately the northern border of Nueces, Jim Wells, Duval, and Webb counties (Figure 4), which roughly corresponds to the 66 °F isotherm (based on the average annual temperature for 1981 to 2010) [30]. I refer to this area of TX south of the 66 °F isotherm as the Rio Grande Valley–Texas Gulf Coast FBT Refugium (Refugium for short). Presumably, this is the area of TX that was most climatically favorable for flea development and infection transmission prior to the 2000s [44]. In the 1990s, 273 out of the 307 (88.9%) of the FBT cases in TX occurred in the Refugium’s 13 counties, which comprise only 6.8% of the area of TX; only 34 cases (11.1%) were reported outside of the Refugium. By contrast, in the decade 2010–2019, 2238 cases (60%) occurred in the Refugium; 1512 cases (40%) were reported outside the Refugium (45-X higher than the 1990s). Thus, in the 2010s, the areas favorable to flea proliferation and infection transmission significantly expanded northward outside the previous most favorable zone. A graph of the number of cases of FBT cases occurring in TX from 1944 to 2019 is shown in Figure 8. From this figure, it is apparent that there was a steep drop in the number of cases when rodent and flea control programs were implemented in the mid-1940s, and then there was a steady low level of cases from 1960–2003 and then a rise in the number of cases in the new millennium.

In the Rio Grande Valley, it is likely that both climatic factors and socioeconomic conditions contribute to the high incidence of FBT. In 2019, the McAllen–Edinburgh–Mission Metropolitan Statistical Area (MSA) was the poorest in the country with a 27% poverty rate. The Brownsville–Harlingen MSA ranked as the fourth poorest with a 25.6% poverty rate [45]. Yao et al. analyzed the spatial distribution of 555 cases of FBT reported from 1996 to 2006 in 18 counties in South Texas [46]. Most FBT cases were seen in urban areas with relatively high human population densities and low household incomes and home values, i.e., areas of lower socioeconomic status (SES). Thus, Yao et al. proposed several demographic factors that contribute to a higher prevalence of FBT in these areas: urban areas provide increased harborage and the availability of pet food and water sources for opossums; and higher numbers of companion cats and dogs and strays in urban areas may act as reservoirs of FBT, blood meal sources for flea proliferation, and mechanical carriers of fleas. Environmental hygiene is likely be lower in low SES areas, resulting in increased harborage and food sources for opossums and stray/feral dogs and cats. Stray domestic animals are known to be more common in urban areas of lower SES [47,48,49]. Also, there are higher rates of flea infestation of pet dogs and cats belonging to owners from lower income areas, which is likely due to less attention to flea control [50].

The epidemiology of FBT in TX in the period 1990–2019 presents an example of infection re-emergence: the reappearance of a known infection after a decline in incidence. Infection re-emergence is often dependent on human influence on ecological systems, causing habitat alteration and changes in species assemblage and contact rates between vectors/hosts and humans. The spread and persistence of re-emerging pathogens can be precipitated by land-use changes, such as urbanization, and by the proliferation of reservoir and vector populations [51]. The re-emergence of a vector-borne infectious disease in a particular geographic area is the result of the responsible pathogen adapting to changes in the vector and/or host ecologies [52].

Urbanization and suburbanization may have multiple effects on the epidemiology of FBT. First, the pet dogs and cats of the populace act as an ongoing source of stray and feral animals that may act as flea hosts and infection reservoirs. Urbanization/suburbanization also alters wildlife communities, generating low biodiversity with increases in the abundance of generalist urban-adapted species, such as opossums, skunks, and raccoons [51,53,54], which are cat flea hosts and reservoirs for flea-transmitted pathogens facilitating parasite spillover into natural areas [55]. A close proximity between natural and urban/suburban habitats increases the exposure of wildlife to feral and domestic animals, which facilitates the *C. felis* infestation of wildlife.

## 6. The Rickettsiae

Flea-borne typhus is caused by *R. typhi* and *R. felis*, obligate intracellular Gram-negative bacteria in the Class Alphaproteobacteria, Order Rickettsiales [56]. *Rickettsia felis* was not definitively identified until 1990, when it was observed in the cytoplasm of midgut cells of cat fleas collected from feral cats and differentiated from *R. typhi* by the polymerase chain reaction (PCR) [57]. In humans, infection with either *R. felis* or *R. typhi* produces mutual serologic cross-reactivity with routine indirect fluorescent antibody testing [58]. Because of this cross-reactivity, it is uncertain if the causative organism for all the prior clinical and epidemiologic studies on FBT was *R. typhi* or *R. felis* or both [59].

Although other pathogenic rickettsiae, such as *R. rickettsii* and *R. prowazekii*, adversely affect the fitness of their arthropod hosts (ticks and the human body louse, respectively), *R. felis* and *R. typhi* are not thought to influence flea longevity and fecundity [60,61,62]. This is due to the ability of these rickettsiae to evade the flea immune response [63] and indicates a long evolutionary association of these bacteria with the flea [64]. *Rickettsia felis* is known to undergo transovarial transmission based on its detection in freshly deposited cat flea eggs [65] and trans-stadial transmission based on observing *R. felis* in newly emerged unfed adult cat fleas [66]. A population of cat fleas can vertically maintain *R. felis* for at least 12 generations without feeding on an infected host, although the infection rates decline from 63% to 2.5% [62]. The host blood source may also influence the sustainability of *R. felis* in cat fleas [62]. The prevalence of *R. felis* within cat fleas also varies between colonies [67]. Other investigators have postulated that the high variability of the vertical transmission of *R. felis* in *C. felis* indicates that this organism does adversely affect flea fitness; thus, some degree of horizontal transmission is necessary for long-term pathogen maintenance [63,68,69,70]. In horizontal transmission, an uninfected flea acquires infection by taking a bloodmeal from a rickettsemic host [5]. For *R. typhi*, the ingestion of only a few rickettsial organisms is sufficient to result in active infection of the flea [71]. The ingested rickettsiae then enter the midgut epithelium, initiating a process of extensive replication [72]. After replication within the midgut cells, rickettsiae are released into the gut lumen and excreted in the feces [3,73]. Traditionally the route of transmission of *R. typhi* from flea to host was thought to be by the posterior station or stercorarian transmission (the inoculation of infected flea feces into cutaneous lesions or mucus membranes [3,69]. Flea feces become infectious approximately 10 days after the fleas acquire the infection, and the fleas remain infectious for the remainder of their lives [73]. Infection of fleas with *R. felis* likely requires a higher level of host bacteremia than for *R. typhi* [68]. The transmission of *R. felis* in the absence of flea feces (i.e., transmission by flea feeding) has been demonstrated by the induction of an antibody response in cats briefly fed upon by *R. felis*-infected cat fleas in which no fecal contamination of the cat’s skin was evident [74]. However, it is uncertain if *R. felis* is transmitted by the posterior station route, although *R. felis* can be found in the feces of infected fleas [4,68,75].

Another mode of transmission is the inhalation of infected flea feces, which may remain infective for up to nine years [5]. More recently, the observation of *R. felis* in the salivary gland of *C. felis* by electron microscopy, PCR, and immunofluorescence suggests the potential for infection of vertebrates through blood feeding [76,77]. A simple mechanical transmission of rickettsiae may also occur because the organism is released from contaminated mouthparts into the skin of the host during probing by the flea [69,78]. Azad and Traub showed *R. typhi* could be transmitted to rats by *X. cheopis* by the oral route [79]. Flea-to-flea transmission of rickettsiae may occur by the process of cofeeding; i.e., when there is simultaneous feeding of multiple fleas on one host, pathogens are transferred from an infected flea to an uninfected flea that is feeding nearby due to the regurgitation of rickettsiae into the bite site by the former. Thus, host rickettsemia is not necessary for flea-to-flea transmission [4]. Flea larvae feeding on *R. felis*-infected feces, flea eggs, or other larvae did not acquire the infection [62].

Multiple studies have documented the simultaneous presence of *R. typhi* and *R. felis* in the cat flea and the Oriental rat flea *X. cheopis* [34,61,80,81]. Cat fleas can also be experimentally infected with both species [82]. However, it is not clearly known if either pathogen has any advantage for acquisition, persistence, or transmission by fleas [81].

In ticks, interference occurs, in which the establishment of one rickettsial species inhibits the transovarial transmission of a second species [83]. However, if *R. felis*-infected cat fleas were fed blood containing *R. typhi* for nine days, cat fleas contained both pathogens; however, infection rates occurred at a lower prevalence than with either single infection, indicating that flea infection with *R. felis* may inhibit *R. typhi* infection [82]. It is unknown if *R. felis* inhibits the vertical transmission of *R. typhi* or transmission to a susceptible vertebrate host [75].

In studies from different locales, the prevalence of *R. felis* infection of cat fleas ranges from 5% to 45.8% [84,85]. Based on PCR studies conducted on fleas, the global distribution of *R. felis* does not necessarily coincide with that of *R. typhi;* the latter is less commonly observed [61,75,86,87,88]. For example, in a study conducted in Orange and Los Angeles Counties (CA) in 2007–2008, about 48% of cat fleas tested positive for *R. felis*, but less than 2% were infected with *R. typhi* [34].

Both species of rickettsiae display “pathogen flexibility” or polyhostality, which is the ability of a pathogen to infect more than one host and more than one order of hosts; data will be presented in part 2 of this series showing that these rickettsiae can infect dogs, cats, and opossums. Polyhostality increases the likelihood that a pathogen will be emergent or re-emergent [54,89,90]. Pathogens that spill over to the human population typically involve a network of species, which is the case for *R. typhi* and *R. felis*. The phylogenetic and ecological host breath of a pathogen increases its zoonotic potential, especially if both domestic and wild hosts are infected [52,91]. Multi-host pathogens, such as *R. typhi* and *R. felis*, are typically associated with multi-host vectors, such as the cat flea [92].

However, some researchers have questioned the pathogenicity and epidemiologic significance of *R. felis* [93,94,95]. In 2016, Billeter and Metzger collected cat fleas from cats in Los Angeles Co. (an endemic area for human FBT) and in Sacramento and Contra Costa Counties (non-endemic areas). PCR confirmed the presence of *R. felis* in cat fleas from both the endemic and non-endemic areas; *R. typhi* was not detected. Because *R. felis* was widespread in cat flea populations in both FBT endemic and non-endemic areas, the investigators concluded that it is unlikely that *R. felis* is a major cause of human FBT in California [93]. Although *R. felis* has been reported as a cause of fever in Africa, it has also been detected from skin swabs from afebrile patients [96], suggesting this organism is a commensal [94]. Nevertheless, over 100 cases of FBT (sometimes designated as flea-borne spotted fever (FBSF) to distinguish it from the disease caused by *R. typhi*) have been specifically ascribed to *R. felis* based on PCR and serologic tests that are able to differentiate *R. felis* and *R. typhi* infections [59,94]. These cases have been documented in the Americas, Asia, Africa, the Pacific, and Europe [97]. Based on limited studies, *R. typhi* is more common in humans in TX [58,98].

### 6.1. Rickettsia typhi; Its Strains, Flea Vectors, and Modes of Transmission

*Rickettsia typhi* belongs to the Typhus Group of the Family Rickettsiaceae. Rickettsiae adhere to endothelial cells with their outer membrane proteins (Omp); OmpA and OmpB are found in the Spotted Fever and the Transitional Groups, whereas Typhus Group rickettsiae lack OmpA [99]. Eight different strains of *R. typhi* have been characterized [100]. *Rickettsia typhi* has been detected in at least twelve species of fleas of nine genera [101]. *Rickettsia typhi* can be ingested by fleas during a blood meal and transmitted vertically to progeny and horizontally to other fleas by cofeeding [3].

### 6.2. Rickettsia felis; Its Transmission Depends on Specific Clades of C. felis

*Rickettsia felis* is now placed in the Transitional Group of the Rickettsiaceae based on the discovery that it possesses a conjugative plasmid [102]. The first reported human case of *R. felis* infection was reported from TX in 1994 [98]. Although *R. felis* has been detected in over 40 species of hematophagous arthropods globally, the only known competent vector is the cat flea [4,65,75]. The cat flea can acquire and horizontally transmit multiple genotypes of *R. felis* [70,84,103,104,105], but how the genotype affects virulence and transmissibility is unknown. Also, the ability of *R. felis* to infect *C. felis* depends on the specific clade of the cat flea [106]. The clinical presentation of *R. felis* infection in humans is similar to that of *R. typhi* [75] but may be milder [97].

## 7. The Cat Flea and Its Many Hosts

The cat flea was discovered as a vector of FBT to humans in 1942 when four persons in Austin, TX, became ill with typhus after exposure to cat fleas on a kitten [107]. However, at that time in the United States, the cat flea’s significance as a vector was overshadowed by the Oriental rat flea. In 1969, the cat flea was incriminated in additional cases of FBT in TX [28], and in 1970, Adams et al. implicated the cat flea in the epidemiology of FBT in the Los Angeles (CA) area [108].

Most species of fleas infest only a single order or family of mammals [109]. Of the 2500 species of fleas [110], only nine species are considered indiscriminate feeders; in North America, these are *C. felis*, *Pulex simulans*, *P. irritans*, *Echidnophaga gallinacea*, and *Nosophyllus fasciatus* [109]. The former four species are known to harbor both *R. typhi* and *R. felis* [3], and *N. fasciatus* is a vector of *R. typhi* [73]. The cat flea possesses genal (head) spines and pronotal (thoracic) combs to facilitate adherence to the fur of its host, but it is not specialized to the pelage of a specific host [109]. Cat fleas feed on a wide diversity of mammals; 138 species have been found to harbor cat fleas. The principal host groups for cat flea infestation are felines (Felidae); foxes and dogs (Canidae); possums and opossums (Phalangeridae, Didelphidae); skunks (Mephitidae); porcupines (Hystricidae); mice and rats (Muridae); shrews (Soricidae); hedgehogs (Erinaceidae); and weasels (Mustelidae). Thus, they are among the most host generalist of all ectoparasites, which is a characteristic that facilitates host switching at the human–wildlife interface [55]. Those host species that frequently enter anthropogenic habitats are at highest risk to acquire cat fleas. However, all hosts are not equally advantageous for flea survival. For example, cat fleas fed on rats consumed more blood, produced more eggs, and had higher offspring female-to-male sex ratios than those fed on mice [111]. Blakey et al. compared dog blood to cat blood and found that cat fleas consume higher quantities of dog blood and produce greater numbers of eggs [112].

In the USA, multiple studies have found that *C. felis* is the most prevalent flea species on dogs, cats, and opossums [113,114]. Other North American wildlife species that harbor *C. felis* include coyotes, red and gray foxes, bobcats, skunks, raccoons, ringtails, ferrets, pumas, and several species of rodents [100,113,114,115]. Disease transmission within an ecological community is more likely when the vector parasitizes a diversity of host species [116]. Habitat encroachment by urbanization and suburbanization increases the potential contact between wildlife and flea-infested human-associated hosts. Thus, anthropogenic habitat use by wild and feral domestic mammals is a strong positive predictor of *C. felis* infestation. The odds of cat flea infestation for anthropogenic habitat-using species is increased by 256% compared to species that do not use anthropogenic habitats [55].

The cat flea serves as a bridging vector between sylvatic hosts (opossums, racoons, skunks), or hosts with intermittent human contact (free-roaming dogs and cats), and domestic dogs and cats [72,73]. The broad host range of the cat flea and the ability of a developing flea to remain in its cocoon until a host is present are both adaptations to survive conditions of irregular host availability [117]. The domestication of cats and dogs by humans has also contributed to cat flea survival by providing favorable locations within human dwellings to complete its life cycle year-round [118].

The proclivity of the cat flea to attack both cats and dogs has led to its widespread geographic distribution [119]. It has been called “the most pervasive flea species on Earth [55]”. The cat flea, being a generalist, is less susceptible to environmental disruptions that may affect the populations of a single host. Host-opportunistic fleas generally display a greater tolerance to climate and environmental variability and thus are more geographically widespread [120,121] and attain higher local densities by exploiting the ability to parasitize multiple hosts [110]. An increased abundance of generalist vectors is strongly associated with increased pathogen transmission [116,122].

In a multi-host pathogen system, the dynamics of the zoonotic agent involves two phases: (i) transmission between maintenance and/or non-maintenance host species (wildlife and/or domestic) and (ii) spillover transmission to humans from the maintenance community. The most important host species communities for FBT in the United States in this case are opossums and both domestic and free-roaming dogs and cats. For FBT, the force of infection from flea to human depends on the prevalence of infection in the flea vector and the rate of contact between humans and the vector (which will depend on the abundance of the vector) [51].

The cat flea meets several criteria of a generalist vector able to successfully exploit the environmental changes wrought by humanity in the 21st century USA: wide geographical distribution; capacity to feed on a range of hosts; capacity for zoophily and anthropophily; ability to exploit peri-domestic and peri-urban settings; and the ability of the larva to utilize indoor and outdoor habitats [123]. The adult flea’s requirement for multiple feedings per day potentially increases its ability to infect multiple hosts [124]. However, the transmission efficiency from flea to host is low (rates of 10–30% for *R. felis* [78], and transovarial transmission is likely the primary mechanism of rickettsial persistence in the flea population. The cat flea also avidly bites humans [125,126]. Thus, the cat flea has emerged as the most important vector of FBT in recent decades in TX and CA [72,115,127].

## 8. Flea-Borne Typhus and Climate Change: General Considerations

Climate change refers to long-term shifts in weather conditions resulting from human activities [128]. Temperatures in TX have increased almost 1.5 °F (0.83 °C) since the beginning of the 20th century. Although there is no overall trend in the number of hot days, the number of warm nights was particularly high during the 2010s. The urban heat island effect has amplified this problem in TX cities [129,130].

Temperature, precipitation, and humidity may affect the reproduction, life cycle, behavior (infestation and biting rates) and population dynamics of arthropod vectors [131]. Thus, climate change has been implicated in affecting the epidemiology of multiple vector-borne infections, including malaria, leishmaniasis, plague, and dengue [132]. Increasing temperatures may increase pathogen transmission by decreasing vector generation time, increasing vector population growth rate, decreasing vector winter mortality, decreasing the pathogen extrinsic incubation period, and lengthening the seasonal duration of vector activity. However, there may also be decreased vector longevity at higher temperatures [19,133]. *Ctenocephalides felis* is already globally widespread; climate change will probably not affect its geographic distribution. However, climate change effects on the cat flea life cycle may increase its density within its established range [119,134].

At many locations throughout its global distribution, FBT tends to be seasonal with the highest incidence rates observed in the warmer months and fewer cases seen in the winter [15,135,136,137]. These results are in accordance with the seasonality of flea infestation, which tends to be higher during warmer months [50,138,139]. For example, in Laos, FBT is more common during the hot, dry years following El Niño [140]. In Taiwan, from 1992 to 2009, a significant correlation was found between average monthly temperature and the number of cases of FBT [141].

Because fleas are poikilothermic, temperature will affect their life histories [142,143]. Weather and climate determine the temperature and relative humidity of the host’s den, which is affected by the ambient temperature, precipitation, and vegetation cover. These conditions will determine the developmental durations and survival of both pre-imaginal and adult fleas [110]. Temperature also greatly affects rickettsial growth within cat fleas; *R. typhi* grows at levels > 1000-fold higher at 24 °C and 30 °C compared to 18 °C [144].

The maintenance cycle of FBT in the environment depends on the thermal tolerance of the bacteria, the flea hosts (opossums, cats, and dogs), and the fleas. If the thermal tolerance for any component of the system is exceeded, the maintenance cycle will be disrupted, and thus the number of cases of human FBT is expected to decrease. However, even during the extreme TX drought and heat wave year of 2011, the number of cases of human FBT increased, indicating that the limits of thermal tolerance of the components of the maintenance cycle were not exceeded even under the extreme weather conditions of 2011. Up to that time, in that year TX had the highest average June-to-August temperatures and the least rainfall since data were first collected in 1895 [145]. Despite these sweltering conditions, TX cases of FBT more than doubled in 2011 to 286 compared to the 135 cases in the previous year [6]. In 2023, TX once again endured record high temperatures and drought conditions. Although the data on FBT cases are not yet available, it will be interesting to see how these extreme conditions affected the epidemiology of FBT for that year.

High levels of rainfall decrease flea infestation, either because the eggs and larvae wash away from the hosts or rain promotes microbial growth that is harmful to the pre-imago stages of the flea life cycle [146]. Conversely, cycles of drought may restrict the distribution of the cat flea in certain areas [147].

All six high FBT prevalence counties showed increases in the overall minimum temperature (Tmin), the mean temperature (Tmean), and the maximum temperature (Tmax) for the decade 2010–2019 compared to 1990–1999, except Travis Co. had a slight decrease in the Tmin (Table 7). Although a less than 1 °F temperature increase may not seem sizeable, this effect was, on average, operative over an entire decade, and temperature affects multiple events in the FBT transmission cycle: rickettsial replication in the flea, infestation rates, biting frequency, flea mating frequency, and the duration of the off-host flea life cycle, as will be described in subsequent sections of this paper. Since climate change effects are often greater at higher latitudes, the temperatures were also obtained for Tarrant Co. (major city Fort Worth) in North Texas, which is a county in which the number of cases of FBT increased from 1 in the 1990s to 93 in 2010–2019. As predicted, Tarrant Co. had a larger increase in the average Tmin, Tmean, and Tmax compared to the other six more southerly high FBT incidence counties.

## 9. The Urban Heat Island Effect in Texas

In the continental USA, urban areas are on average 2.9 °C warmer than surrounding areas due to replacement of the natural soil and vegetation with artificial surfaces [148,149]. There are reduced daily temperature fluctuations due to higher nocturnal temperatures in cities compared to the countryside [150]. The urban heat island effect is complex and depends on vegetation cover, seasonal effects, and proximity to bodies of water. In Dallas from 2001 to 2011, the July average was 2.4 °C higher than the surrounding rural area [151]. In San Antonio, from June 1 to September 30 during the years 2002 to 2008, the nocturnal temperature was 6–7 °C higher than the surrounding area [130]. Urban sprawl amplifies the heat island effect. For example, the Austin metro area population increased from 655,000 in 1993 to 1.43 million in 2011 (US census data); in that period, the average surface temperature of Austin rose by 4.7 °C [152]. The urban heat island effect has definite biological consequences; for example, there is an increase in the vegetation growing season by about 15 days in urban areas relative to adjacent rural areas [153]. Thus, the urban heat island effect may prolong the period of flea activity during cooler months and at night.

## 10. The Cat Flea Life History and the Effects of Temperature and Humidity: General Aspects

To understand how climate change may affect the incidence of FBT in TX, it is necessary to consider the effects of temperature on each stage of the cat flea life history: host seeking and infestation, feeding, mating, oviposition, hatching of the eggs into larvae; feeding by the larvae; pupation; and eclosion. In terms of its life history, the cat flea can be described as a fur flea (versus a nest or fur/nest flea). Fur fleas spend more time on the host and thus are potentially exposed to higher fluctuations in ambient temperature and humidity compared to nest fleas [121]. Gracia et al. have proposed that *C. felis* has a wider environmental tolerance than *C. canis* or *P. irritans* [154].

### 10.1. Host Seeking and Infestation: The Effect of Temperature

After emergence from its cocoon, the imperative for a newly emerged adult flea is to locate a host. Higher temperatures promote jumping activity in fleas, which is necessary for host acquisition [110]. Although the direct transfer of fleas from one host to another does occur, the more common route of infestation is the acquisition of newly emerged fleas from the environment [113]. Data from various locales indicate that cat flea infestation of cats, dogs, and opossums is more common during warmer months [50,80,113,138,139,154,155,156,157,158,159,160,161,162,163,164,165,166]. A specific example of the effect of temperature on flea infestation was seen in Germany during a heat wave in 2003 in which the prevalence of flea infestations of dogs and cats doubled [138]. In a Spanish study from 2002 to 2004, a positive relationship was found between *C. felis* abundance on dogs and the mean annual temperature without effects of rainfall [154]. Thus, warmer weather due to climate change may increase the prevalence and intensity of flea infestation [119].

Female and male fleas typically take their first blood meal within two hours of emerging from their cocoons and then feed 4–10 times per day thereafter. There is a specific temperature threshold that must be met to stimulate the initial feeding [167]. For males, a bloodmeal is essential to initiate mating behavior [168]. Furthermore, the epididymis of the newly emerged unfed male cat flea is obstructed by folded columnar epithelium, but the initial blood meal unblocks the epididymis, permitting sperm transfer [169].

The feces of adult fleas constitute the major component of the larval diet. The maggot-like larvae also consume flea eggs and injured flea larvae [170]. At higher temperatures, there is a higher rate of flea feeding and defecation [110]. Also, the rate of digestion is accelerated at higher temperatures and is associated with lower energy expenditure for digestion, allowing fleas to allocate more metabolic resources to other activities, such as mating or egg production [171].

An increased frequency and duration of feeding may increase the probability that a flea will ingest infected blood during a period of host bacteremia [172] or increase the likelihood of acquiring infection from another flea by cofeeding. Farhang et al. found higher fecal output from cat fleas that fed ad libitum on rats at 24 °C and 30 °C versus 18 °C [144]. Increased defecation increases the amount of flea feces in the flea nursery, providing nutriture for larval growth and development. Larger larvae are less prone to desiccation due a smaller surface-to-volume ratio [173]. Thus, if larvae are better fed because of a more bountiful fecal food supply, they may be more likely to survive. Furthermore, a well-fed larva produces an adult with higher fat reserves better able to withstand a period without access to a blood meal [110].

### 10.2. Mating and Oviposition: The Effect of Temperature

Feeding by adult fleas, induced by warmer temperatures, is a stimulus for mating [174]. Temperature also determines mating frequency in cat fleas. Under experimental conditions, at 27 °C, no mating occurred. Between 34 °C and 42 °C, mating commenced with most of copulation occurring at 38 °C. However, there were no mating attempts at 44 °C [168]. Male cat fleas may inseminate multiple females within 24 h after a blood meal. Multiply mated female cat fleas have higher fecundity (total eggs) and fertility (egg viability) than singly mated females because with each mating, the female may acquire nutrients, oviposition stimulants, and viable sperm, all of which enhance their reproductive success [174]. Because *R. felis* may be transmitted sexually between fleas [63], increased copulation frequency and copulation between different pairs of fleas may also promote the transmission of rickettsiae between fleas. Virgin female cat fleas lay nonviable eggs after a blood meal; however, within 24 h after male fleas are introduced, they began ovipositing viable eggs and quadruple their egg output [175].

Female fleas decrease egg production during colder weather and resume egg laying when environmental conditions become favorable for larval development [176]. The number of eggs produced by the adult female flea correlates with volume of blood consumed [168]. Larger females (presumably from feeding more frequently) have higher lifetime fecundity [64,177]. It has been proposed that male fleas with larger body size have increased mating success [171]. Adult flea longevity likely increases with larger body size [178], allowing fleas to perform a greater number of matings during their lifetimes [174]. A female cat flea produces about one egg an hour while on the host. The eggs readily fall from the host’s fur, accumulating in areas where the host spends the most time [113,179].

### 10.3. Egg Hatching and Larval Development; Effects of Temperature and Humidity

The duration of the developmental period and the survival of preimaginal fleas are highly dependent on microclimatic conditions [180]. Thus, flea abundance is strongly influenced by off-host environmental conditions [110], because only 1–5% of the flea population (all stages) live on their host at a given time with the remainder dispersed around the resting or feeding areas of the host [181].

The eggs of *C. felis* hatch within 1.5 to 6 days following oviposition. Temperature and humidity determine egg survival and the incubation period. At 35 °C and 70% relative humidity, most eggs hatch within 36 h; at 13 °C, egg hatching requires six days [182]. Optimally, the flea larvae development occurs in protected microenvironments that combine moderate to warm temperatures, high humidity, and an ample supply of protein-rich flea feces and infertile eggs for larval nutriture [183]. The development of the eggs and larvae is restricted by temperatures outside a range of 4–35 °C and by a relative humidity < 50% [184]. The high FBT-incidence cities of TX all have sufficient humidity for optimal flea larval development. The average relative humidity levels of Brownsville, Houston, Corpus Christi, San Antonio, and Austin are 90%, 89%, 89%, 83% and 83%, respectively [185].

An underground burrow is ideal as a flea nursery because it is warmer in the winter and cooler and more humid during hot weather. The larvae develop through three stadia, and the survival and rate of development of each stage depends on the temperature and humidity. The duration of the larval stage is 5–11 days, depending on the temperature, humidity, and food availability [182]. The highest rate of emergence (82%) of adult cat fleas occurs when the larvae have access to feces and nonviable eggs compared to feces alone (47%); feeding on nonviable eggs also shortens the duration of larval development [174]. The number of nonviable eggs depends on mating frequency, which is temperature-dependent [168]. The larval stage is highly sensitive to low humidity due to the inability of flea larvae to absorb atmospheric water at low humidity levels and to withstand respiratory water loss [182,186].

### 10.4. The Temperature-Driven Infestation ⟶ Feeding ⟶ Reproduction ⟶ Larval Trophic Cascade

Thus, higher environmental temperatures for newly emerged adult fleas promote an infestation ⟶ feeding ⟶ reproduction ⟶ larval trophic cascade. After eclosion, increased temperature promotes active host questing by the newly emerged adult flea. Once a host is acquired, higher temperatures stimulate feeding. Feeding and higher temperatures stimulate mating, which leads to the production of both viable and nonviable eggs with the latter serving as a food source for larvae. Increased feeding leads to larger adult body size, which is associated with higher fecundity. The more frequent feeding due to higher temperatures leads to greater adult flea feces production. A greater availability of feces and nonviable eggs in the flea nursery promotes accelerated larval development and increased fitness.

### 10.5. Pupation and Pupa to Adult Eclosion

The pupal stage is less sensitive to temperature than the larval stage with its duration equal from 16 to 27 °C. However, a temperature of 8 °C for 20-days is lethal as well as >35 °C [187]. During unfavorable conditions (too cold, hot, or dry), cocoons enter diapause until environmental circumstances improve [120]. The ability to survive for extended periods in the cocoon is important for species such as *C. felis*, which infest mobile hosts that may not frequently return to the same nest or burrow. Increasing temperatures and mechanical pressure stimulate emergence of the adult [188]. The pupae are the most environmentally-resistant stage of the life cycle and can survive for up to six months awaiting a host [167].

### 10.6. Overall Temperature Effects on the Cat Flea Life Cycle

Thus, at higher temperatures, fleas progress through their life cycles at a faster rate, i.e., eggs hatch faster, and the duration of the larval stage of fleas is reduced. In one study conducted in north central Florida (a latitude comparable to South TX), fleas survived all year outdoors. In June and July, eggs developed into adults in 20–24 days, whereas in the winter, it took 36–50 days. The immature stages survive frosts in protected microhabitats. From September to November, larval survival was up to 85% [173].

## 11. Endosymbionts of the Cat Flea: Possible Temperature Effects

*Steinina ctenocephali* is a temperature-dependent endosymbiotic protozoan of the cat flea alimentary canal. Flea larvae infected with *S. ctenocephali* develop faster than an uninfected control group [189]. Fleas were more likely to be infected with this protozoan during warmer seasons [190].

Bacteria of the genus *Wolbachia* are endosymbionts of insects that may affect their reproduction, longevity, and vector competence and efficiency [106]. The infection of insects by *Wolbachia* may also affect their thermotolerance and thermal preferences [191], which may be relevant to rickettsial replication in the flea and flea behavior. Three *Wolbachia* strains have been isolated from *C. felis* [106]. It is uncertain how *Wolbachia* may affect *R. typhi*/*R. felis* acquisition, maintenance, and/or transmission in *C. felis* [106,192], so this is a question that needs to be investigated in areas of high FBT incidence. In a study of cat fleas from Malaysia, Tay found that *R. felis* was found only in *Wolbachia*-positive fleas [193]. *Wolbachia* endosymbiosis of insects displays a complex temperature dependence [194].

## 12. The One Health Model of Flea-Borne Typhus Epidemiology in Texas in the Early 21st Century: The Importance of Human Population Growth and Increasing Temperatures

Figure 9 summarizes the One Health Model of FBT epidemiology in TX in the early 21st century. In the lower right corner, human population growth leads to increasing urbanization and suburbanization. Human population growth causes a warming climate along with an urban heat island effect. Warmer weather promotes the flea infestation of host mammals, feeding, and defecation. It also accelerates the flea life cycle and increases rickettsial replication with the flea. Increased rickettsial replication facilitates the transmission of rickettsiae to host animals and humans. Human population growth also indirectly increases the population of dogs and cats, both domestic and free-roaming, that can act as flea hosts and possibly as reservoirs. Opossum populations also increase in an urban environment [12]. Opossums will enter yards and pass fleas to dogs and cats, which, in turn, may spread infected fleas into human domiciles. The control of opossum and stray dog and cat populations and the use of flea control products on pets can serve to suppress the transmission of FBT to humans.

## 13. Limitations of the Data and Alternate Explanations for the Rise in Typhus Cases

The reliance on serologic testing for the diagnosis of FBT has a number of pitfalls that likely result in an underestimation of the true number of cases of FBT in TX. Anti-rickettsial antibodies are present in less than 20% of patients at seven days after the onset of illness [95]; it may take up to 14 days for antibodies to develop, and confirmation of infection requires a 4-fold rise of antibody titers between acute and convalescent periods [195]. If patients are lost to follow-up evaluation after their acute illness, a confirmatory convalescent serologic test will not be obtained. Also, convalescent testing may not be performed if the medical practitioner seen at a follow-up appointment deems that the test is no longer important or if the patient does not have the means to afford the test. Cases of FBT are identified only after a patient seeks health care, a serologic diagnosis is made, and the data are captured by the reporting of serologic tests; thus, mild cases that did not come to medical attention or cases treated empirically without serologic testing being performed will be missed. Also, lack of health insurance and healthcare access likely reduces the recording of cases among medically underserved populations [15]. Furthermore, there are gaps in the passive surveillance system in which clinical cases are appropriately recorded at the state level. A 2017 study of pediatric patients with FBT in Houston found that only 48% of the 31 confirmed or probable cases were reported to the TDSHS [196].

Are there alternate explanations for the rise in the number of cases of FBT in TX in the new millennium, such as greater recognition of the infection or improved diagnostic capabilities? In the TX Department of State Health Services study of 2003–2013, the investigators doubted that greater case ascertainment was the cause of the increase in the number of reported cases [31], but they did not provide an explanation. There has not been a concerted effort by public health authorities to broadly educate physicians or the public about FBT in the past twenty years. Furthermore, the main diagnostic method, the indirect fluorescent antibody test, has been used since 1976 [7]. This test still requires a reference laboratory in the 2010s as it did in the 1990s, so ease of diagnosis has also not improved.

## 14. The Control of Flea-Borne Typhus

As discussed, the main proposed drivers of the increased number of FBT cases in TX are human population growth and climate change. These inexorable forces cannot be altered in the short-term. Nevertheless, there are actions that can be taken on an individual and community-wide basis that decrease the risk of infection. Krueger and coworkers of the Orange County Mosquito and Vector Control District (CA) have outlined in detail the components of a FBT control program based on the reduction in flea and host density and the modification of human behavior to reduce flea exposure [197]. The public must be educated about the dangers of fleas on pets, strays, and urban wildlife. The reduction in flea density in the household requires that pet owners carry out flea control measures throughout the year. Furthermore, urban wildlife must be discouraged from feeding and nesting in residential and commercial areas. This would include rodent control methods (rodenticides/trapping) and xeriscaping/hardscaping to limit harborage sites and food for urban wildlife, rodents, and free-roaming cats and dogs. Xeriscaping/hard scaping also limits habitat for flea life cycle completion. Other environmental management strategies include removing potential animal harborage and trimming fruit trees and removing fallen fruit that may entice urban wildlife into areas near human habitation [197]. Flea-borne typhus control intersects with general measures to reduce stray/feral dog and cat populations. These actions include promoting the sterilization of pets when they are acquired and enforcing laws that limit the number of pets per household and the feeding of free-roaming animals [197]. Many communities have implemented Trap–Neuter–Release programs to manage free-roaming cats. However, unless a minimum of 70% sterilization rate of the free-roaming cats is achieved, these programs do not effectively reduce cat populations. The supplemental feeding of colonies of free-roaming cats may attract urban wildlife, creating the potential for the interspecies exchange of fleas and pathogens between animals and transmission to humans [59]. A robust public health response to outbreaks of FBT in a community is necessary to determine the source of infection in order to decrease additional human cases. In outbreak situations, areas inhabited by cat colonies and other sites of high flea infestation may require the local application of insecticides [197].

## 15. Research Questions About FBT in Texas

The schema proposed in Figure 9 facilitates the generation of research questions to further understand the current epidemiology of FBT in TX. Understanding the dynamic relationships of each component to each other will allow the generation of predictive models for the future epidemiology of FBT. Specific research questions are listed below:

#1 What is the true magnitude of the stray and feral cat and dog problem in the high FBT incidence areas of TX and their degree of flea infestation over the course of a year?

#2 How do the numbers of opossums, dogs, and cats and strays relate to increases in human cases in individual counties?

#3 What is the net effect of climate change on the flea life cycle (is the decrease in life cycle duration countervailed by the decrease in adult flea life span)?

#4 Are changes in temperature, rainfall, and relative humidity associated with increased human cases in specific locales? Does a statistically significant relationship exist between these climatic variables and the number of human cases?

#5 What endosymbiotic organisms (*Wolbachia pipientis* and *Steinina ctenocephali*) occur within the flea vectors in TX, and how do they affect flea survival and infection with *R. felis* and *R. typhi*?

#6 What is the impact of co-infection of *R. felis*/*R. typhi* and *Bartonella* sp. on flea survival and vectorial capacity?

#7 Are *R. felis* and *R. typhi* still co-circulating in TX? Do they vary geographically and by season?

## 16. Conclusions

Flea-borne typhus is an infection caused by generalist bacteria and spread by generalist fleas with flea hosts that are also generalists. This increases the risk of spread of FBT to pets and humans. It is difficult to control from a public health perspective, because its transmission is closely associated with human behavior, domestic pets, stray/feral dogs and cats, and urban wildlife [59]. Human population growth in TX promotes FBT by the increased urbanization and abundance of pet dogs and cats, stray/feral dogs and cats, and opossums. Increasing temperatures in Texas in the new millennium increases the flea-borne transmission of typhus by promoting host infestation and flea feeding and defecation, accelerating the flea life cycle, and increasing rickettsial replication within the flea. Increased opossums and stray cats and dogs in the urban/suburban landscape increase the risk of flea transfer to humans and their pets.

## Figures and Tables

**Figure 1 pathogens-14-00154-f001:**
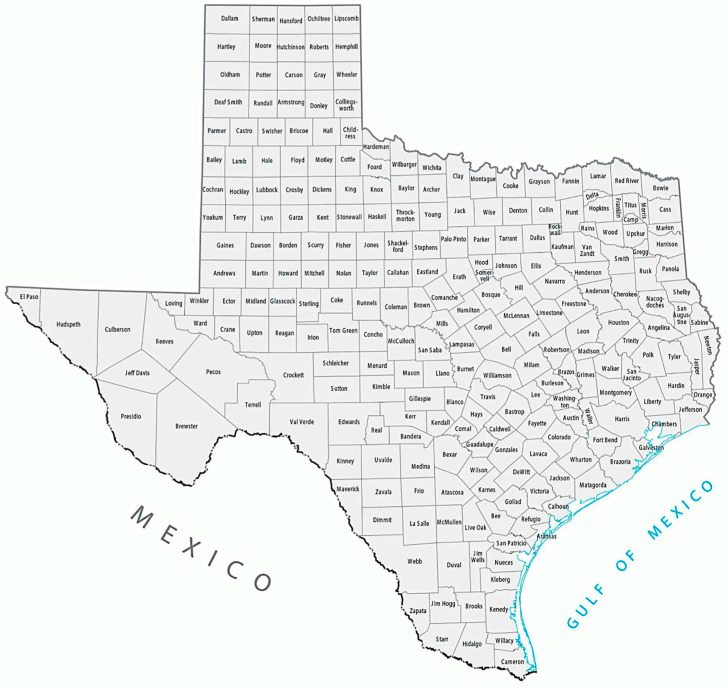
Map of the 254 counties of Texas.

**Figure 2 pathogens-14-00154-f002:**
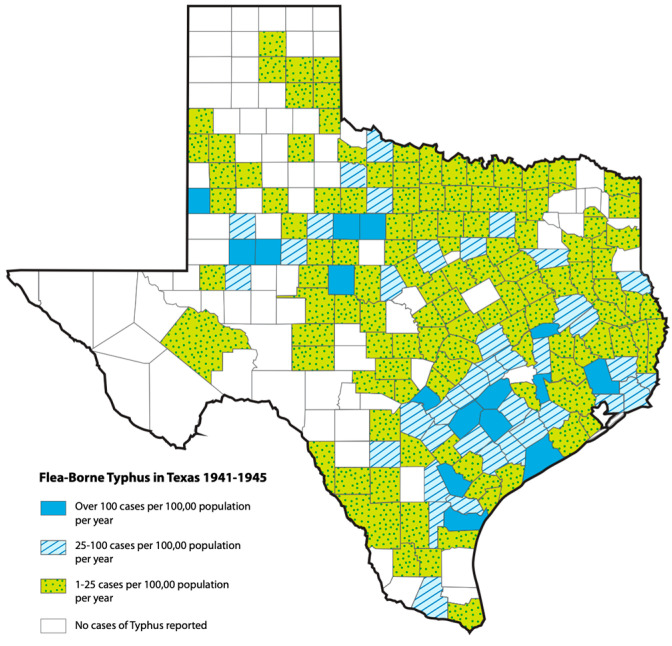
Cumulative incidence of flea-borne in the counties of Texas, 1941–1945 [27]. The 16 highest incidence counties were Nueces, Yoakum, Martin, Howard, Runnels, Jones, Shackelford, Bee, Gonzales, Dewitt, Lavaca, Comal, Matagordo, Waller, Liberty, and Madison (see Figure 1).

**Figure 3 pathogens-14-00154-f003:**
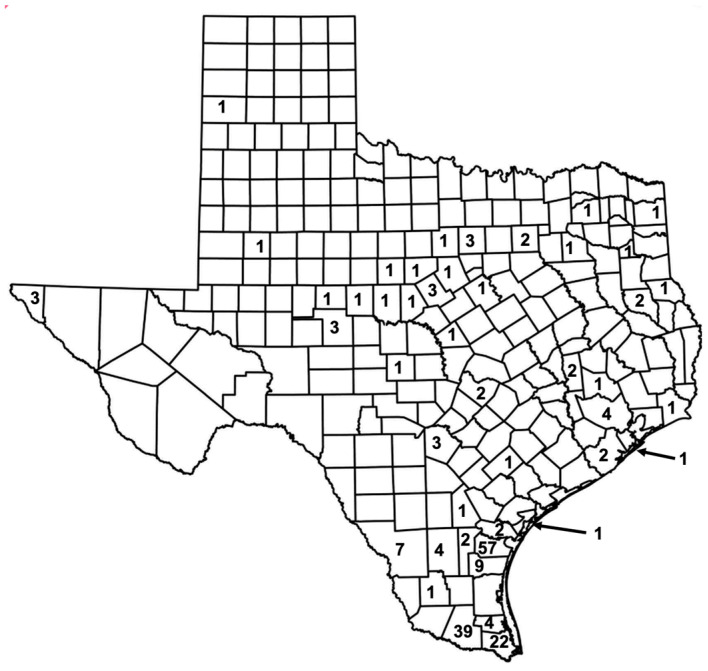
Geodistribution of flea-borne typhus cases in Texas, 1980–1984 [29].

**Figure 4 pathogens-14-00154-f004:**
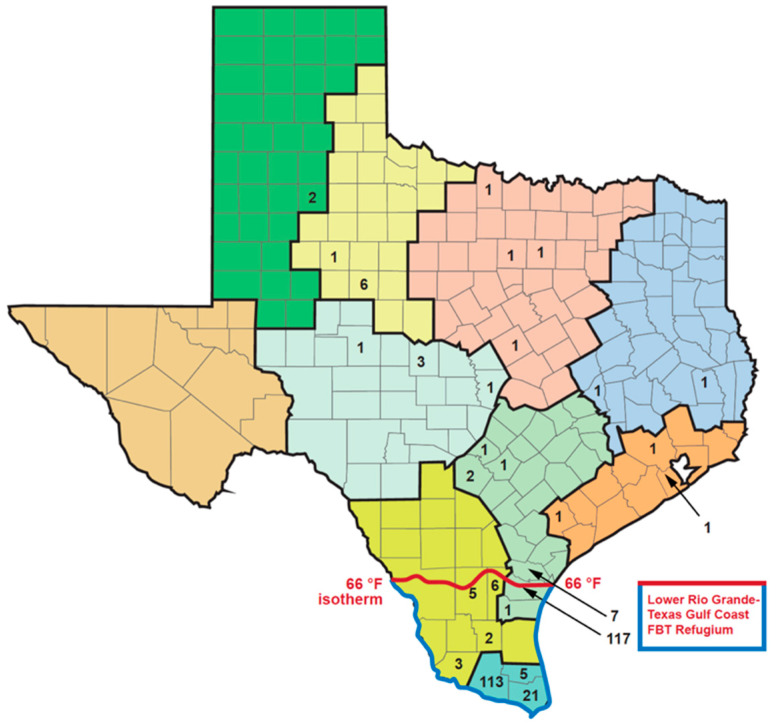
Geodistribution of the 307 cases of flea-borne typhus by county in Texas 1990–1999 [9]. There were 28 counties with cases; 88.9% of cases occurred in the Refugium’s 13 counties with only 34 cases outside the refugium (11.1%). Another seven cases occurred in San Patricio Co., a county bordering the Refugium, leaving only 27 cases (8.8%) outside the Refugium and a Refugium bordering county. The different colors represent specific National Climatic Data Center climate zones [30], which will be described in detail below.

**Figure 5 pathogens-14-00154-f005:**
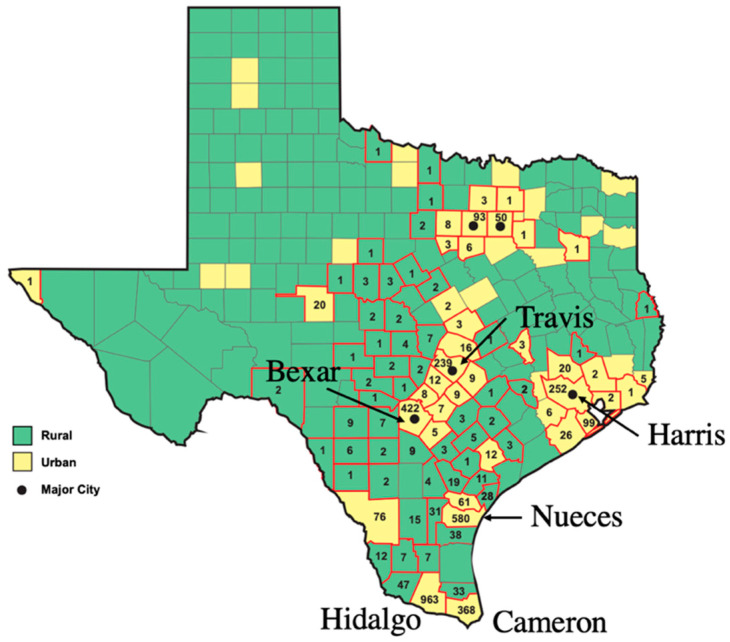
Geodistribution of cases of flea-borne typhus in Texas in 2010–2019 based on the urban or rural classification of each county, naming the six urban counties with the highest number of FBT cases.

**Figure 6 pathogens-14-00154-f006:**
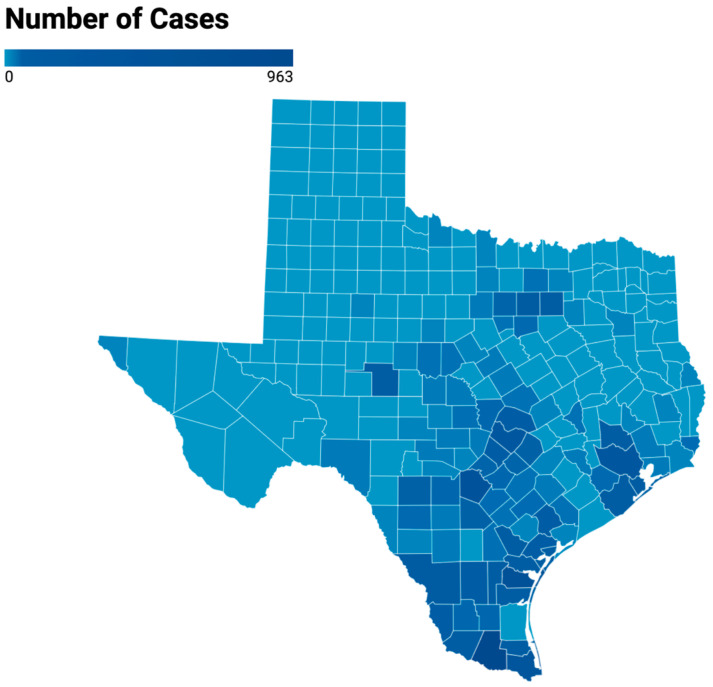
Choropleth map of the number of cases of flea-borne typhus in each county of Texas during 2010–2019.

**Figure 7 pathogens-14-00154-f007:**
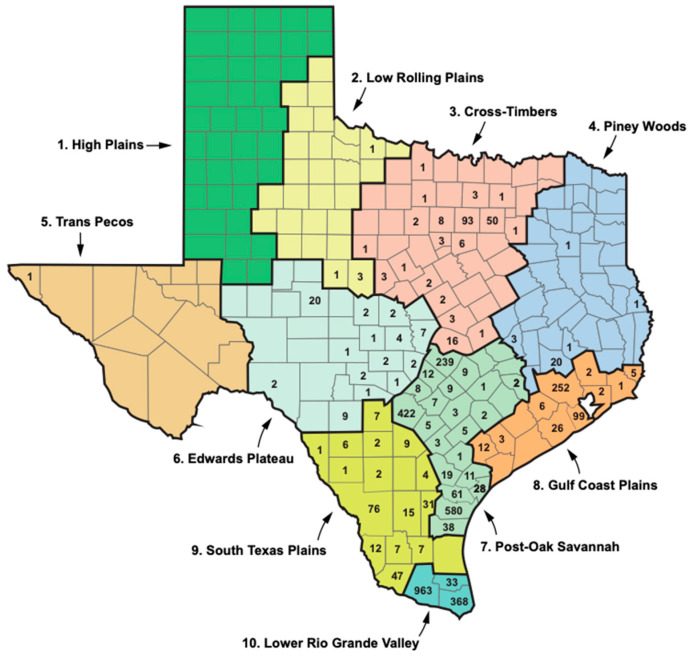
The geodistribution of FBT cases in Texas with respect to the ten National Climatic Data Center climate zones with each color representing a different climate zone (see Table 6) [30], 2010–2019.

**Figure 8 pathogens-14-00154-f008:**
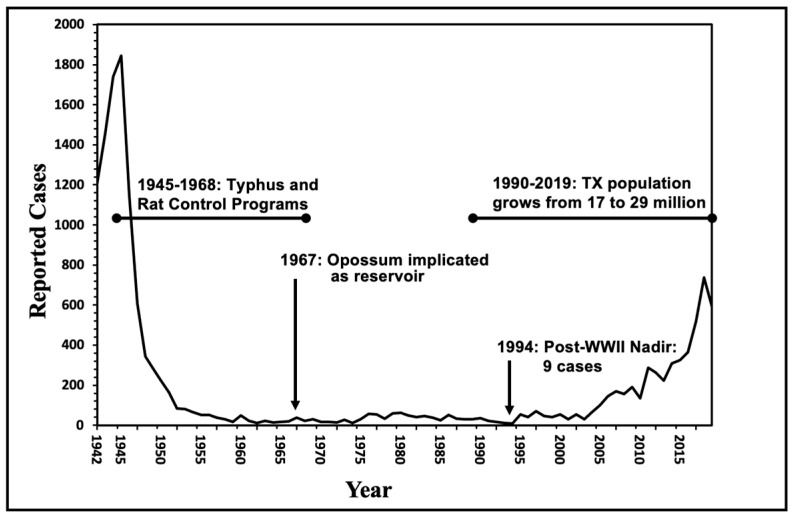
Epidemiologic curve of flea-borne typhus in Texas, 1942–2019 [6,7,9].

**Figure 9 pathogens-14-00154-f009:**
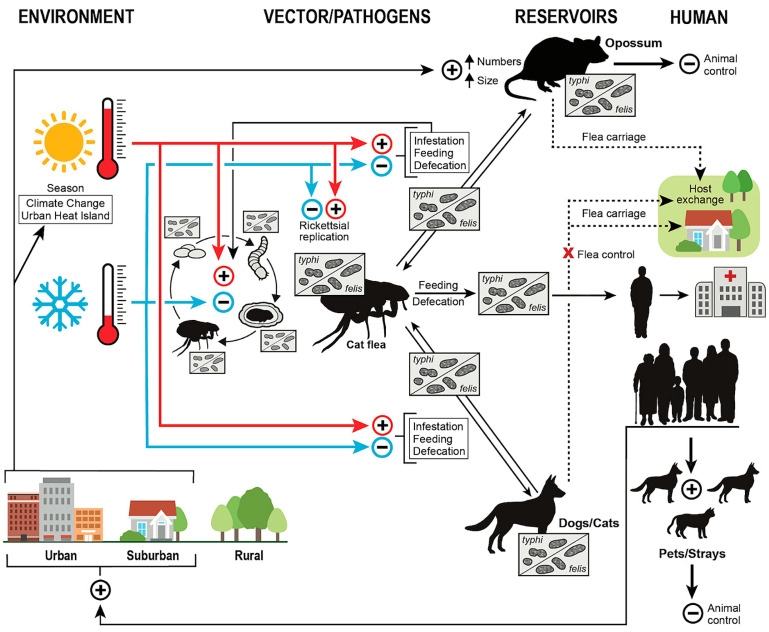
One Health model of flea-borne typhus epidemiology in Texas in the early 21st century. The dotted lines indicate flea carriage into areas of human habitation.

**Table 1 pathogens-14-00154-t001:** Abiotic, biotic, and anthropogenic factors of the episystem of flea-borne typhus in Texas in the 20th and 21st Centuries.

Abiotic	Biotic	Anthropogenic
Temperature	Abundance of urban exploiter hosts ^a^	Population growth
Rainfall	Abundance of companion animals	Urbanization/suburbanization
Relative humidity	Abundance of other wildlife hosts	Agricultural practices
	Prevalence of flea endosymbionts ^b^	Human effects on climate
	Cat flea clade-*Rickettsia* compatibility	Urban heat island effect
		Pet ownership
		Control of urban exploiter hosts
		Flea control: pets/environmental
		Environmental sanitation
		Housing conditions

^a^ Opossums, rats, and stray/feral dogs and cats. ^b^ *Wolbachia pipientis*, *Steinina ctenocephali*.

**Table 2 pathogens-14-00154-t002:** Population growth of Texas and Texas counties with the highest number of flea-borne typhus cases, 1990 versus 2019 ^a^.

	1990 ^a^	2019 ^a^	Percent Increase, 1990 to 2019
Texas	16,986,510	28,995,881	70.7
Metropolitan (% of total)	14,165,650 (83.4)	25,920,625 (89.4)	83.0
non-metro	2,820,852	3,075,261 ^b^	9.0
Bexar	1,185,394	2,004,000	67.1
Cameron	260,120	423,163	62.7
Harris	2,833,000	4,713,000	66.3
Hidalgo	383,545	868,707	126
Nueces	291,145	362,294	24.4
Travis	576,407	1,274,000	121

^a^ From US Census data except where indicated. ^b^ [32].

**Table 3 pathogens-14-00154-t003:** Increase in average annual number of cases of FBT in Texas and the six highest incidence counties, 1990–1999 versus 2010–2019.

State or County	Largest City	Total Number of Cases 2010–2019 ^a^	Avg Number of Cases/Years, 1990–1999 ^b^	Avg Number of Cases/Years, 2010–2019 ^a^	Fold Increase in Avg No. of Cases, 2010–2019 vs. 1990–1999
Texas	-----	3750	30.7	375.0	12.2
Bexar	San Antonio	422	0.2	42.2	211
Cameron	Brownsville	368	2.1	36.8	17.5
Harris	Houston	252	0.1	25.2	252
Hidalgo	McAllen	963	11.3	96.3	8.5
Nueces	Corpus Christi	580	11.7	58.0	5.0
Travis	Austin	239	0	23.9	---

^a^ [6]; ^b^ [9].

**Table 4 pathogens-14-00154-t004:** Estimated pet and stray dog populations in the six Texas counties with highest prevalence of flea-borne typhus, 1990–1999 and 2010–2019.

County	Avg. Human Pop, 1994/1995 ^a^	Est. Pet Dog Pop., 1994/1995 ^b^	Est. Stray Dog Pop., 1994/1995 ^c^	Est. Total Dog Pop., 1994/1995	Avg. Human Pop. 2014/2015 ^a^	Est. Pet Dog Pop. 2014/2015	Est. Stray Dog Pop, 2014/2015 ^c^	Est. Total Dog Pop, 2014/2015	Increase in Total Dog Pop. 2014/2015 vs. 1994/1995
Bexar	1,280,000	345,600	75,294	420,894	1,876,500	506,655	110,382	617,037	196,143
Cameron	287,204	77,545	16,894	94,439	420,705	113,590	24,747	138,337	43,898
Harris	3,050,000	823,500	179,412	1,002,912	4,506,000	1,216,620	265,059	1,481,679	478,767
Hidalgo	468,443	126,480	27,555	154,035	868,707	234,551	51,100	285,651	131,616
Nueces	310,079	83,721	18,240	101,961	362,294	97,819	21,311	119,130	17,169
Travis	656,303	177,202	38,606	215,808	1,274,000	343,980	74,941	418,921	203,113

^a^ US Census data. ^b^ Human population X 0.27 [35]. The human population of the decade midpoint is used. ^c^ Based on 17:1 human to stray dog ratio [37]. Abbreviations: Avg., average; Est., estimated; Pop., population.

**Table 5 pathogens-14-00154-t005:** Estimated pet and stray cat populations in the six Texas counties with highest prevalence of flea-borne typhus, 1990–1999 and 2010–2019.

County	Avg. Human Pop, 1994/1995 ^a^	Est. Pet Cat Pop., 1994/ 1995 ^b^	Est. Stray Cat Pop., 1994/1995 ^c^	Est. Total Cat Pop., 1994/1995	Avg. Human Pop. 2014/2015 ^a^	Est. Pet Cat Pop. 2014/2015	Est. Stray Cat Pop, 2014/2015 ^c^	Est. Total Cat Pop, 2014/2015	Increase in Total Cat Pop. 2014/2015 vs. 1994/1995
Bexar	1,280,000	268,800	107,520	376,320	1,876,500	394,065	157,626	551,691	175,371
Cameron	287,204	60,313	24,125	84,438	420,705	88,348	35,339	123,687	39,249
Harris	3,050,000	640,500	256,000	896,500	4,506,000	946,260	378,504	1,324,764	455,264
Hidalgo	468,443	98,373	39,349	137,722	868,707	182,428	72,971	255,399	117,677
Nueces	310,079	65,117	26,046	91,163	362,294	76,082	30,432	106,515	15,352
Travis	656,303	137,823	55,129	192.953	1,274,000	267,540	107,016	374,556	181,603

^a^ US Census data. ^b^ Human population X 0.21 [35]. The human population of the decade midpoint is used. ^c^ Based on stray and feral cats comprising 40% of the pet cat population [38]. Abbreviations: Avg., average; Est., estimated; Pop., population.

**Table 6 pathogens-14-00154-t006:** The number of cases of flea-borne typhus in each of the climatic regions of Texas, 1990–1999 and 2010–2019 ^a,b^.

Region Number ^a^	Region Name/Alternate Name ^c^	Description ^c^	Number of FBT Cases, 1990s ^b^	% of Total	Number of FBT Cases, 2010s ^a^	% of Total	Fold-Increase
1	High Plains	Continental steppe or semi-arid savanna	2	0.7	0	0	Decreased
2	Low Rolling Plains	Sub-tropical steppe or semi-arid savanna	7	2.3	5	0.13	Decreased 0.7
3	Cross Timbers/ North Central	Subtropical subhumid mixed savanna, woodlands	4	1.3	198	5.28	49.5
4	Piney Woods/ East Texas	Subtropical humid mixed evergreen–deciduous forestland	2	0.7	26	0.69	13
5	Trans Pecos	Subtropical arid desert	0	0	1	0.027	------
6	Edwards Plateau	Subtropical steppe or semi-arid brushland, savanna	5	1.6	56	1.49	11.2
7	Post-Oak Savannah/ South Central	Subtropical subhumid mixed prairie, savanna, woodlands	129	41.3	1465	39.1	11.5
8	Gulf Coastal Plains/ Upper Coast	Subtropical humid marine prairies and marshes	3	1.0	408	10.88	136
9	South Texas Plains/Southern	Subtropical steppe or semi-arid brushland	16	5.2	227	6.05	14.2
10	Lower Rio Grande Valley/Lower Valley	Subtropical sub-humid marine	139	45.2	1364	36.37	9.8
Totals	--------	-------	307	86.5% in regions 7 and 10	3750	75.5% in regions 7 and 10	12.2
Refugium ^d^	------	-------	273 (88.9%)		2238 (59.7%)		8.2
Outside Refugium	-------	--------	34 (11.1%)		1512 (40.3%)		44.5

^a^ [9]; ^b^ [6] ^c^ [30]; ^d^ Refugium refers to the 13 counties mostly south of the 1981–2010 66 °F isotherm (see Figure 4).

**Table 7 pathogens-14-00154-t007:** Minimum, mean, and maximum temperatures (°F) in Texas counties with the highest prevalence of flea-borne typhus cases, 1990–1999 versus 2010–2019 ^a^.

County	Tmin, °F			Tmean, °F			Tmax, °F		
	1990s	2010s	Diff	1990s	2010s	Diff	1990s	2010s	Diff
Bexar	58.93	59.43	+0.5	69.60	70.18	+0.58	80.27	80.93	+0.66
Cameron	65.25	66.2	+0.95	74.23	75.18	+0.95	83.19	84.13	+0.94
Harris	60.59	61.9	+1.31	69.80	70.86	+1.06	79.04	79.8	+0.76
Hidalgo	63.98	64.61	+0.63	74.73	75.51	+0.78	85.46	86.37	+0.91
Nueces	63.89	64.65	+0.76	72.33	72.91	+0.58	80.78	81.18	+0.40
Travis	58.51	58.33	−0.18	68.84	69.47	+0.63	79.83	80.62	+0.79
Tarrant	55.67	57.72	+2.05	65.90	67.64	+1.74	76.60	77.57	+0.97

^a^ Data from Prism Climate Group [13].

## Data Availability

The data are contained within the article.

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
