# Peer review of "A One Health Perspective on the Resurgence of Flea-Borne Typhus in Texas in the 21st Century: Part 1: The Bacteria, the Cat Flea, Urbanization, and Climate Change"

_pathogens, 2025, doi:10.3390/pathogens14020154_

Round 1
Reviewer 1 Report
Comments and Suggestions for Authors
The article provides an integrative One Health perspective considering the human, animal and environmental factors in FBT epidemiology and is validated with county-level epidemiological data from Texas. There are however areas for improvement:
The role of urbanization is well explored, but the effects of climate on rural health systems remain unexplored.
The article does not adequately discuss alternative explanations for the increase in FBT cases, such as improved diagnostic capabilities.
Socioeconomic disparities are mentioned, but mechanisms linking poverty and disease transmission (e.g. pet flea control practices, housing quality) are not elaborated.
A limitation that should be mentioned is that trends are described, but statistical models to robustly analyze correlations or causations with climate variables are not employed. This could be mentioned in the research questions.
The narrative is highly detailed, but may overwhelm readers, especially non-experts. It would be good if additional structure is employed in the manuscript. 2 Suggestions:
- Table 1 gives a nice overview of factors relating to FBT epidemiology: the structure of the article could incorporate these parameters with subtitles indicating which factor is discussed. It should also be mentioned if on certain factors (e.g. rainfall or humidity) data/knowledge are lacking.
- Line 527 - line 530: it would be nice if the impact of each of the factors mentioned here, are discussed in a structured way with their effect on FBT.
A geographic heat map pasted next to an incidence map could simplify and highlight key findings.
A short paragraph with actionable proposals would be welcome.
Other specific comments:
Table 1: ‘Climate’ is too vague in the abiotic group. Components of climate are temperature, humidity, rainfall, wind,…
The article is lengthy: although interesting, the historical epidemiology can be shortened because the study addresses the period 1990-1999 and 2010-2019
Figure 6: explain that the colors represent the National Climatic data center zones
Line 373 is repetition of line 362
Line 338 - line 447: interesting paragraph about with general information about transmission, different hosts, and intrinsic properties of R. felis, R. typhi and C. felis, but the link with the manuscript subject One Health is not very clear.
There are several small typographical errors: a spelling check is advised.
Author Response
Comment#1
The role of urbanization is well explored, but the effects of climate on rural health systems remain unexplored. I have focused on the urban areas because 91% of the FBT cases in the period 2010-2019 occurred in urban counties.
Comment#2
The article does not adequately discuss alternative explanations for the increase in FBT cases, such as improved diagnostic capabilities. I have now added a comment about other possible explanations in section 12. "Are there alternate explanations for the rise in the number of cases of FBT in TX in the new millennium, such as greater recognition of the infection or improved diagnostic capabilities? In the TX Department of State Health Services study of 2003-2013, the investigators doubted that greater case ascertainment was the cause of the increase in the number of reported cases, but they did not provide an explanation. There has not been a concerted effort by public health authorities to broadly educate physicians or the public about FBT in the past twenty years. Furthermore, the main diagnostic method, the indirect fluorescent antibody test, has been used since 1976. This test still requires a reference laboratory in the 2010s as it did in the 1990s, so ease of diagnosis has also not improved."
Comment#3
Socioeconomic disparities are mentioned, but mechanisms linking poverty and disease transmission (e.g. pet flea control practices, housing quality) are not elaborated. I have added a comment, supported by three references, about how stray domestic animals are more common in urban areas of lower socioeconomic status. I have added a comment on the relationship between low income and flea infestation of pets. "Stray domestic animals are known to be more common in urban areas with lower socioeconomic status. Also,there are higher rates of flea infestation of pet dogs and cats belonging to owners from lower income areas, likely due to less attention to the flea control."
Comment#5
A limitation that should be mentioned is that trends are described, but statistical models to robustly analyze correlations or causations with climate variables are not employed. This could be mentioned in the research questions. This has now been added to questions #4: "Are changes in temperature, rainfall, and relative humidity associated with increased human cases in specific locales? Does a statistically significant relationship exist between these climatic variables and the number of human cases?"
Comment#6
The narrative is highly detailed, but may overwhelm readers, especially non-experts. It would be good if additional structure is employed in the manuscript. 2 Suggestions:
- Table 1 gives a nice overview of factors relating to FBT epidemiology: the structure of the article could incorporate these parameters with subtitles indicating which factor is discussed. It should also be mentioned if on certain factors (e.g. rainfall or humidity) data/knowledge are lacking.
- Line 527 - line 530: it would be nice if the impact of each of the factors mentioned here, are discussed in a structured way with their effect on FBT. I have added additional text to the titles of the sections to indicate what factors are discussed.
Comment#7
A geographic heat map pasted next to an incidence map could simplify and highlight key findings.
A choropleth map of the FBT cases in each county has been added (new Figure 6). This map shows clusters of counties with higher numbers of FBT cases.
A short paragraph with actionable proposals would be welcome. A section on FBT control has been added.
Comment#8
Other specific comments:
Table 1: ‘Climate’ is too vague in the abiotic group. Components of climate are temperature, humidity, rainfall, wind,…
I have omitted the word climate and left three components of climate that are known to influence the flea life history: temperature, rainfall, and humidity.The reviewer states wind should be included. Wind may be important for winged vectors,such as mosquitos and sandflies, but it is not important for apterous vectors such as fleas.
Comment #9
The article is lengthy: although interesting, the historical epidemiology can be shortened because the study addresses the period 1990-1999 and 2010-2019. The historical section has been shortened. The historical section provides a prelude to understanding the epidemiology in the 1990s and some of the factors listed in Table 1.
Comment #10
Figure 6: explain that the colors represent the National Climatic data center zones. The captions of new Figures 4 and 7 now indicate that the colors refer to the NCDC climate zones.
Comment #11
Line 373 is repetition of line 362: These lines in the original manuscript are not repetitions; perhaps the reviewer meant two other lines?
Comment #12
Line 338 - line 447: interesting paragraph about with general information about transmission, different hosts, and intrinsic properties of R. felis, R. typhi and C. felis, but the link with the manuscript subject One Health is not very clear. I have removed original lines 383-386 on the pathogenesis of R. typhi infection because this is not directly relevant to the One Health model. Many papers on "murine typhus" do not clearly differentiate that there are two potential species involved and so this is why the text on the two species was included. It is also necessary to describe the transmission dynamics and the involvement of pets, free-roaming cats and dogs, and wildlife reservoirs.
Comment t#13
There are several small typographical errors: a spelling check is advised. Spell check was performed.
Reviewer 2 Report
Comments and Suggestions for Authors
Dear Authors,
Here are my review and comments on your manuscript.
The manuscript is well-written and provides valuable insights. However, due to the extensive information it covers, I found it necessary to read it several times to fully understand and follow the content. One aspect I found somewhat challenging was the time shifts across different sections, as some periods overlapped. For instance, Section 4 (Typhus in Texas in the New Millennium: The 2003-2013 Texas DSHS Study) and Section 5 (Building a One Health Model: The Epidemiology of Typhus in Texas, 2010-2019) jump back and forth in time. While the data is well-described, this organization makes the narrative harder to follow.
I suggest presenting the data in chronological order, starting with earlier years and proceeding sequentially to newer years, to enhance the manuscript’s readability and logical flow.
Aside from this, the manuscript is highly informative and clearly outlines an effective approach to studying and mitigating the spread of infections.
I have a few additional comments regarding formatting and one of the figures:
Formatting: There are inconsistent spaces between words throughout the manuscript and tables. This might be due to not using the left-align formatting option, which should be straightforward to resolve.
Figure 7: This figure could benefit from improvements. Currently, it appears disorganized, and the text does not align well with the graph. I recommend reorganizing the figure by presenting data chronologically by year and possibly adding a line or indicator to point out the respective years for clarity.
Thank you for your work on this important topic. I believe the manuscript is a significant contribution to the field.
Author Response
Comment#1
The manuscript is well-written and provides valuable insights. However, due to the extensive information it covers, I found it necessary to read it several times to fully understand and follow the content. One aspect I found somewhat challenging was the time shifts across different sections, as some periods overlapped. For instance, Section 4 (Typhus in Texas in the New Millennium: The 2003-2013 Texas DSHS Study) and Section 5 (Building a One Health Model: The Epidemiology of Typhus in Texas, 2010-2019) jump back and forth in time. While the data is well-described, this organization makes the narrative harder to follow.
Response # 1The sections do not jump back and forth, they overlap. The overIap is unavoidable because the authors of the DSHS studied examined data that fell within the decade that I am examining. I have now described in the text that the The 2003-2013 Texas DSHS Study was the first study to indicate an increasing number of FBT cases were occurring in Texas over the course of their study period. However, the authors were unable to provide an explanation for this situation. The current study re-examines the Texas data with respect to demographics, urbanization, climate change, and the potential of climate change on the flea life history to provide its One Health perspective.
Comment#2 Formatting: There are inconsistent spaces between words throughout the manuscript and tables. This might be due to not using the left-align formatting option, which should be straightforward to resolve.
Response#2
I submitted a manuscript that was not in the template format because I was unable to get the template to behave for me. Formatting errors occurred when the manuscript was placed in the Pathogens template. I have now rectified these and the I am sure the journal's formatting editor will also help for the final version.
Comment #3
(old) Figure 7: This figure could benefit from improvements. Currently, it appears disorganized, and the text does not align well with the graph. I recommend reorganizing the figure by presenting data chronologically by year and possibly adding a line or indicator to point out the respective years for clarity.
Response #3
I have simplified this Figure and the caption.
Reviewer 3 Report
Comments and Suggestions for Authors
Your analysis of flea-borne typhus in Texas, including recent changes in epidemiology, in the context of One Health is very interesting and you make your case well that changes in microbes, vectors and the environment (including changes in human population ) all play a role in the changes observed. Its a very interesting paper and well written, which I enjoyed reviewing.
My comments below are all relatively minor.
1. In the Introduction (lines 37-38) I suggest you add a comment that FBT an be transmitted to humans via inhalation of an infectious aerosol of flea faeces.
2. lines 83-84. Could the increase in FBT cases from 1923 to 1933 be due to greater medical awareness and better lab diagnoses rather than changes in land use by Texas farmers ?
3. line 307 "..control of pets in (not "is") area of ...
4. line 345. "cat fleas is able to vertically..."
5. line 381. "bite site by the..."(space needed)
6. line 467. "Thus they are among the most..."
7. Figure 8 is complex and impressive in its detail. However, the humans shown (the single person and the group of people) both seem to fall partially under the heading of "Reservoirs", which of course they are not. I suggest that these figure be moved slightly to the right so they fall completely under the "Human" heading.
Author Response
My comments below are all relatively minor.
Comment #1
In the Introduction (lines 37-38) I suggest you add a comment that FBT an be transmitted to humans via inhalation of an infectious aerosol of flea faeces.
Response #1 Ok, this has been added.
Comment#2 Lines 83-84. Could the increase in FBT cases from 1923 to 1933 be due to greater medical awareness and better lab diagnoses rather than changes in land use by Texas farmers ?
Response#2 Yes, that is likely so (as now indicated in the manuscript). However, I suggest that the big increases that occurred during the WWII years were due to a transition from cotton to peanuts.
Comment#3 Lline 307 "..control of pets in (not "is") area of ...
Response#3 Corrected
Comment #4: line 345. "cat fleas is able to vertically..."
Response#4 Corrected
Comment #5 line 381. "bite site by the..."(space needed)
Response #4 Corrected
Comment #5
Line 467. "Thus they are among the most..." Response#5
Corrected
Comment #6
Figure 8 is complex and impressive in its detail. However, the humans shown (the single person and the group of people) both seem to fall partially under the heading of "Reservoirs", which of course they are not. I suggest that these figure be moved slightly to the right so they fall completely under the "Human" heading.
Response #6
The Figure (new Figure 9) has been modified as suggested by the reviewer.